# Archaean continental crust formed from mafic cumulates

**Matthijs A. Smit** [1,2] ✉, **Kira A. Musiyachenko**[1] **& Jeroen Goumans**[1]

Large swaths of juvenile crust with tonalite-trondhjemite-granodiorite (TTG) composition were added to the continental crust from about 3.5 billion years ago. Although TTG magmatism marked a pivotal step in early crustal growth and cratonisation, the petrogenetic processes, tectonic setting and sources of TTGs are not well known. Here, we investigate the composition and petrogenesis of Archaean TTGs using high field-strength-element systematics. The Nb concentrations and Ti anomalies of TTGs show the overwhelming effects of amphibole and plagioclase fractionation and permit constraints on the composition of primary TTG melts. These melts are relatively incompatible element-poor and characterised by variably high La/Sm, Sm/Yb and Sr/Y, and positive Eu anomalies. Differences in these parameters are not indicative of melting depth, but instead track differences in the degree of melting and fractional crystallisation. Primary TTGs formed by the melting of rutile- and garnet-bearing plagioclase-cumulate rocks that resided in proto-continental roots. The partial melting of these rocks is part of a causal chain that links TTG magmatism to the formation of sanukitoids and K-rich granites. Together, these processes explain the growth and differentiation of the continental crust during the Archaean without requiring external forcing such as meteorite impact or the start of global plate tectonics.

The rise of tonalite-trondhjemite-granodiorite (TTG) crust during the Archaean[1] is generally considered paramount to the development and growth of the "modern" andesitic continental crust and its habitable environment[2–5]. This foundational step in Earth's evolution went at the expense of ancient proto-continental crust, which developed during the Hadean to Paleoarchaean and has since been largely removed from the rock record[3]. Although Archaean TTGs are central to crustal evolution and cratonisation, there is substantial uncertainty about how these rocks were made. In general, the petrogenesis of TTGs is considered to be a two-stage process, involving (1) the isolation of mafic (tholeiitic) sources that were hydrous and enriched in large ion lithophile elements and (2) the melting of these sources at different depths, as indicated by La/Yb, Sr/Y, and $Eu_N/Eu^{*}$[6–10]—where Eu* is $\sqrt{(Sm_N \times Gd_N)}$. "High-pressure TTGs" with fractionated REE compositions and high Sr/Y could represent melts from early equivalents of subduction zones[5].

Consistent with this interpretation, experiments showed that such TTGs can indeed be produced by garnet-stable and plagioclase-absent melting of basaltic crust[11,12]. The rise of TTG crust with juvenile radiogenic isotope compositions during the course of the Archaean thus has been considered a marker for the onset of modern-style plate tectonics[13–15]. The analogy between TTGs and modern arc granitoids is nevertheless disputed[16]; TTGs may equally have formed through the melting of underplated or over-accreted lower crust, the lower domains of tectonically thickened or plateau-like crust, the roots of thickened arc-like crust, or crustal material caught in (sub-)vertical crustal downwellings[3,8,9,16–26]. Thermodynamic modelling provides new opportunities to test these models and has allowed fundamental reinterpretations of various chemical features. For instance, high La/Yb and Sr/Y values, which were typically attributed to deep garnet-stable and plagioclase-absent melting (≥2.0 GPa[7]), were shown to be equally

¹Department of Earth, Ocean and Atmospheric Sciences, University of British Columbia, 2020-2207 Main Mall, Vancouver V6T 1Z4, Canada. ²Department of Geosciences, Swedish Museum of Natural History, Frescativägen 40, SE-104 05, Stockholm, Sweden. ✉e-mail: msmit@eoas.ubc.ca

possible for TTGs formed at significantly lower pressure (1.4 GPa)[27]. Plagioclase-absent melting, as also indicated by high $Eu_N/Eu^*$ values, neither requires nor implies large melting depth; pressures as low as 1.0 GPa may suffice depending on source composition, redox conditions, and water content[28]. Closed-system, continuous melting and crystallisation to produce TTGs have also been questioned, with alternative models highlighting the importance of mineral segregation[29], loss of interstitial liquids[26], and hybridisation of melts formed at progressively increasing depth[30]. The source rocks provide an additional factor of uncertainty in TTG petrogenesis. Hydrous basaltic rocks are generally considered[8,11,12,23,24] and phase equilibrium modelling confirms that rocks with a hydrous mid-ocean ridge basalt (MORB) composition could indeed melt to form tonalite or trondhjemite melt[31]. Whether the amphibolite rocks that are observed as sources for TTGs in the field[32] were indeed basaltic is difficult to determine given their extensive deformation, metamorphism and melt loss. Some TTG source rocks display cumulate textures, which could indicate a gabbroic rather than basaltic protolith[33]. Alternative source rocks, which could equally explain the compositional features of TTGs, include bimineralic eclogites formed by the interaction between mafic lower crust and picritic melts[34], and hybridised peridotites residing in the metasomatised lithospheric mantle[35].

Insights into the petrogenesis of Archaean TTGs require empirical constraints on the composition of primary TTG melts. Such constraints are difficult to obtain, because most, if not all, TTGs underwent high degrees of crystal fractionation[36,37] and may additionally have accumulated peritectic and wall rock-derived material[26,29]. The chemical effects of these processes may be difficult to disentangle from the compositional features that characterise the melting process. To overcome these challenges and to determine the composition of the parental melts of Archaean TTGs, it is necessary to develop a reliable means to quantify the effects of these various processes. The high

field-strength elements (HFSE) Ti and Nb are of particular interest in this regard[9,20], as they are: (1) relatively fluid-immobile, (2) generally incompatible, yet differently so for different silicate phases, (3) sequestered in the residual titaniferous phases (e.g., rutile, ilmenite) that are stable at different pressure and temperature (P–T) conditions during melting of mafic rocks, and (4) are more abundant then chemically similar elements (e.g., Ta). In this study, fractionation monitors based on HFSE were explored, tested, and applied to obtain insights into the formation of Archaean TTGs.

The initial composition and chemical evolution of TTGs was investigated by focussing on Nb concentration and the Ti anomaly $Ti_N/Ti^*$—where N refers to normalisation to Primitive Mantle[38] and $Ti^* = \sqrt{(Zr_N \times Gd_N)}$. The basic premise of using these parameters is that, during melting, residual titaniferous phases lower $Ti_N/Ti^*$ in the melt, but affect Nb concentrations of the melt differently depending on which phase is involved. Melting to form TTGs likely occurs at P–T conditions of 1.0–1.8 GPa and 800–950 °C[39]. Experiments[40] and phase equilibrium modelling[31,41,42] show that titanite is not stable in mafic systems under such conditions. Titanite may occur in Ca-rich basaltic rocks that could compositionally represent TTG source rocks but is lost when such rocks are heated beyond ca. 750 °C[31,42]. Rutile or ilmenite thus is expected in the residue, depending on whether melting occurs at the high (rt) or low end (ilm) of the P–T range[41,43]. In basaltic systems, the partition coefficient for Nb in rutile ($D_{Nb}^{rt}$) is ca. 30 times higher than $D_{Nb}^{ilm}$ (ca. 1.5; Supplementary Data 1). Whereas $D_{Nb}^{rt}$ is high enough to counterbalance the incompatibility of Nb in other residual silicates during melting (e.g., garnet, clinopyroxene), $D_{Nb}^{ilm}$ is not. Thus, although melts derived from a given mafic source containing titaniferous phases will always evolve to lower $Ti_N/Ti^*$, they will become poorer or richer in Nb, depending on whether rutile or ilmenite is present among the residue (Fig. 1b). Assimilation of material containing rutile or illmenite, either in the source or elsewhere in the

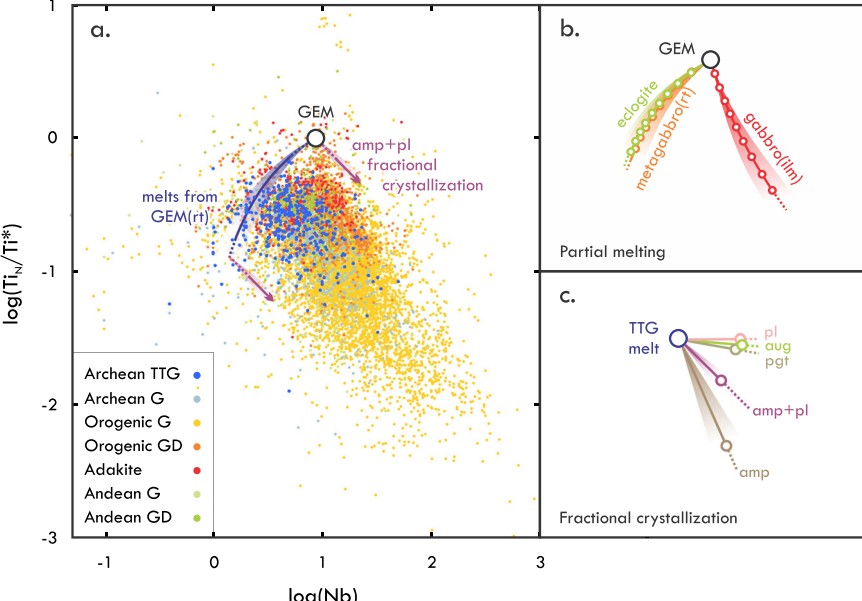

**Fig. 1 | Nb–Ti systematics of TTGs. a** The Ti anomaly versus Nb concentration for granitoids. The curves track compositions expected for melts derived from a GEM source comprising rutile-bearing lithologies ("GEM(rt)"), which are similar for eclogite ($cpx_{45}amp_{10}grt_{40}rt_5$) or rutile-bearing metagabbro ($cpx_{40}grt_{25}pl_{30}rt_5$). The general compositional change in melts caused by the fractional crystallisation of amphibole and plagioclase (1:1) is shown for reference. Orogenic granite (G) and granodoirite (GD) are from collisional orogens and other orogenic settings, and Andean G and GD are from Andean margins worldwide. Data for each granitoid type are provided separately in Fig. S1. **b** Melt compositions expected for a GEM made of eclogite, rutile-bearing metagabbro, and ilmenite-bearing gabbro

($cpx_{40}grt_{25}pl_{30}ilm_5$; "gabbro(ilm)"). The vectors are to scale compared to (**a**). Markers indicate melt compositions at 10% melting intervals, with lower degrees of melting being compositionally more distinct from GEM. **c** Compositional changes in TTG melts undergoing fractional crystallisation of common silicates (and a 1:1 assemblage of amphibole and plagioclase). Markers show the change after 50% fractional crystallisation (scale twice that of **a**). Background information, including a description of reference granitoids and figures for individual granitoid types, is provided in Supplementary Note 1 and calculations are provided in the "Methods" section.

magmatic plumbing system, would have the opposite effect on Nb and $Ti_N/Ti^*$. The effect from the assimilation of titaniferous phases far outweighs that of any silicate minerals (e.g., clinopyroxene, garnet, amphibole, plagioclase), because these are relatively poor in Nb and the $D_{Ti}$ is comparable to $D_{Gd}$ for these phases during melting of a mafic source (Supplementary Data 2). Amphibole would have the largest effect, as it may contain substantial HFSE and has $D_{Ti} > D_{Gd} > 1 > D_{Zr}$. However, more than 70% of amphibole from a granitoid wall rock would need to be assimilated to raise $Ti_N/Ti^*$ of a TTG melt by a factor of 2. This would produce a hornblendite rather than a TTG. Fractional crystallisation could have a more profound effect than assimilation. Melt modelling of andesitic-dacitic TTG analogues shows that the loss of common saturating phases (e.g., clinopyroxene, amphibole, plagioclase) drives melt compositions to higher Nb along vectors that differ depending on phases (direction) and the degree of crystallisation (magnitude; Fig. 1c). Amphibole fractionation has by far the largest effect due to its high $D_{Ti}$ (Supplementary Data 2). Re-melting of granitoids will cause further fractionation along similar mineral-controlled chemical vectors. From these systematics, it can be predicted that the rocks with the lowest Nb and highest $(Ti_N/Ti^*)/Nb$ among genetically related granitoids would approximate most closely the melt from which these rocks ultimately derive. These concepts were used to evaluate the composition of global granitoids and to build a reference frame for TTGs (Supplementary Note 1). This evaluation shows that the above predictions are reflected in the global granitoid record.

## Results and discussion
### Composition of primary TTG melts

The compositions of Archaean TTGs[44] are anchored at high $Ti_N/Ti^*$ by a Granitoid End Member (GEM in terms of Nb–$Ti_N/Ti^*$, similar to average continental crust; Supplementary Note 1) and are bracketed at low Nb by the high $(Ti_N/Ti^*)/Nb$ residual-rutile trend that is also seen in modern granitoids (Fig. 1a, S1b). Ilmenite can be ruled out as a residual phase of significance on the basis of these observations. If ilmenite had been a common component of the source, TTGs should generally have been more Nb-rich than GEM. Concomitant depletion in both Ti and Nb, as seen in high-$(Ti_N/Ti^*)/Nb$ TTGs, would not be possible. This would apply even more so for low-Ti mafic sources that contain magnetite or titanomagnetite instead of ilmenite because these would deplete the melt far less in Ti. The Nb-$(Ti_N/Ti^*)$ systematics instead indicate that: (1) TTG petrogenesis involved the melting of either a common source (GEM) with residual rutile or of sources that differentiated from GEM in the presence of rutile, and (2) TTG melts start Nb-poor with high $(Ti_N/Ti^*)/Nb$ and evolve to more Nb-rich compositions during their magmatic differentiation. Those TTGs with the highest $(Ti_N/Ti^*)/Nb$ likely represent primary melt compositions, because the only common process by which TTG melts can become depleted in Nb—the accumulation of late plagioclase segregations[29]—can be ruled out as generally controlling chemical differences. Such accumulation would lead to predict that high-$(Ti_N/Ti^*)/Nb$ TTGs are relatively CaO-rich, which is not the case. The high-$(Ti_N/Ti^*)/Nb$ TTGs with the lowest Nb, which require the largest amount of plagioclase accumulation to explain their trace-element signatures, in fact have the lowest CaO concentrations (ca. 2 wt%; Supplementary Note 1). If high-$(Ti_N/Ti^*)/Nb$ TTGs are primary melts from a common source, then they may capture an igneous differentiation trend and this is indeed the case; the TTGs show progressively higher $SiO_2$, and lower FeO, MgO, $TiO_2$, CaO and $P_2O_5$ with decreasing $Ti_N/Ti^*$ (Supplementary Note 1). This trend shows that primary TTGs formed by the melting of a common rutile-bearing GEM source at different degrees, with relatively mafic melts (ca. 62 wt% $SiO_2$, ca. 2 wt% MgO and ca. 6 wt% of CaO) and felsic melts (ca. 75 wt% $SiO_2$, <0.2 wt% MgO and ca. 1 wt% CaO) produced at high and low degrees of melting, respectively. The TTGs evolve to more Nb-rich compositions mainly as a result of fractional crystallisation and possibly the assimilation of evolved crustal

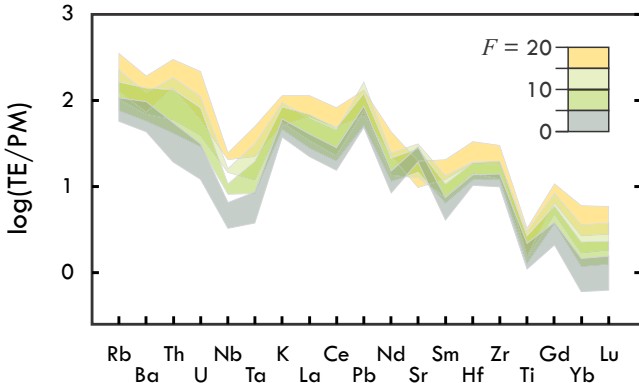

**Fig. 2 | Trace-element (TE) composition of TTGs as a function of fractionation factor $F$.** With increasing $F$, TTGs show higher incompatible-element concentrations, except for Sr and Pb. TTG with $F > 20$ extend compositional trends, but are few in number and hence not shown. Data are normalised to Primitive Mantle[38].

materials. The net enrichment in Nb of the melt due to these processes can be parameterised through the fractionation factor $F$ (see the section "Methods"). The main advantage of this factor is that it provides a relative measure of the cumulative effect of magmatic differentiation without requiring assumptions regarding the initial $Ti_N/Ti^*$ of the melt or the degree of melting.

The TTGs with low $F$, which reflect the lowest degree of fractional crystallisation, have low incompatible-element concentrations, large Sr and Pb excesses and highest La/Sm, Sm/Yb, Sr/Y, and $Eu_N/Eu^*$ values (Figs. 2 and 3). The fractionated REE compositions set these rocks apart from adakites and other convergent-margin granitoids, all of which show near-unity $Eu_N/Eu^*$, and La/Sm and Sm/Yb values that resemble those of MORB[37] (Fig. 3). This supports the argument made on the basis of insignificant mantle–wedge interaction that TTGs and adakites are not mutual analogs[16]. With increasing $F$, TTGs show higher concentrations of REE, lower Sr and Pb excesses, higher Ba/K, and lower La/Sm, Sm/Yb, $Eu_N/Eu^*$, K/La and Ba/Th (Fig. 2). These changes are consistent with the fractional crystallisation of amphibole and plagioclase. Amphibole fractionation drives down Ba/Th of the melt and impacts HFSE budgets, raising Nb and lowering $Ti_N/Ti^*$ (Fig. 1c). Fractional crystallisation of plagioclase, and sanidine in more evolved stages of melt evolution[37], progressively lowers K/La, Sr/Y, $Eu_N/Eu^*$ and Ba/Th of the melt[29], while raising Ba/K. Although the REE are generally incompatible in amphibole and plagioclase, the heavy REE (HREE; Er–Lu) and the middle REE (MREE; Sm–Ho) other than Eu are more incompatible in plagioclase than light REE (LREE; La–Nd) are in amphibole. The MREE are slightly more compatible than HREE in amphibole, whereas they are uniformly incompatible in plagioclase. The net effect of the fractional crystallisation of amphibole and plagioclase thus is the observed REE enrichment at decreasing $Eu_N/Eu^*$ and the progressive loss of high La/Sm and Sm/Yb values (Fig. 3), the latter possibly aided by melt mixing. The observed trends support the major role of amphibole and feldspar crystal fractionation in TTG composition[26,29,37], and are consistent with the generally low modal abundance of amphibole in TTGs[8] and observations of hornblende cumulates[36] and plagioclase segregations[29] with compositions complementary to TTGs. The compositional evolution of TTGs appears uniform during the Archaean. The most extreme La/Sm, Sm/Yb, and $Eu_N/Eu^*$ are observed for Meso- and Neoarchaean TTGs, but elevated values also sporadically occur among older TTGs. Instead of indicating a secular change in the petrogenetic processes and setting of TTG magmatism[24,36], apparent changes in the REE signatures of TTGs may reflect preservation bias, with younger TTGs being better-preserved and analysed more frequently resulting in an higher

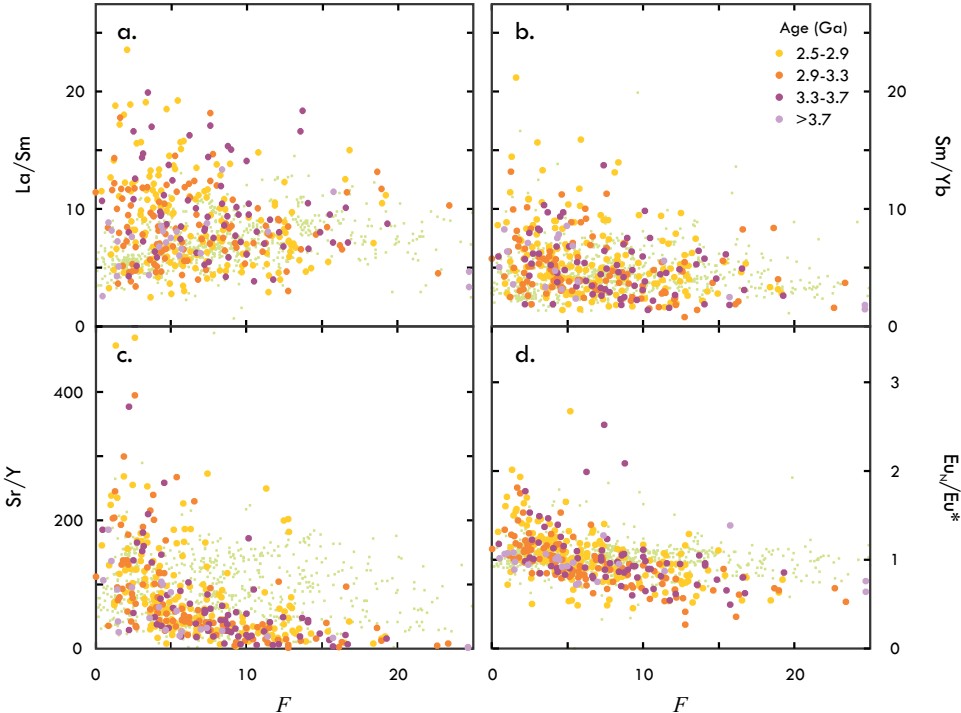

**Fig. 3 | The REE composition of TTGs as a function of fractionation factor $F$.** Primitive TTGs with limited effects from crystal fractionation show higher **a** La/Sm, **b** Sm/Yb, **c** Sr/Y and **d** $Eu_N/Eu^*$ regardless of their age. Adakite compositions (green) are shown for reference.

apparent incidence of unfractionated, low-$F$ TTGs with variably high La/Sm and Sm/Yb.

## Mafic plagioclase-cumulate source rocks

The large range in La/Sm and Sm/Yb as seen in primary TTGs with low $F$ would conventionally be interpreted as indicating a range in melting depth[8]. However, all of these TTGs, including those with low La/Sm and Sm/Yb, have high $(Ti_N/Ti^*)$/Nb indicative of deep rutile-stable melting ($P > 1.4$ GPa[31]). This adds to the conclusion that factors other than melting depth control REE ratios[27]. Positive Eu anomalies in TTGs have been ascribed to amphibole fractionation on the basis of an inverse correlation between $Eu_N/Eu^*$ and Yb[36]. The inverse correlation between $Eu_N/Eu^*$ and $F$, however, shows that this interpretation is incorrect and that high $Eu_N/Eu^*$ is a primary feature of TTGs. High La/Sm, Sm/Yb and $Eu_N/Eu^*$ are particularly pronounced among primary TTGs with low $Ti_N/Ti^*$ and MgO (Supplementary Note 1), which could indicate that differences in these REE signatures−like differences in major element concentrations−stem from differences in the degree of melting. Modal melt modelling (see the "Methods" section), however, indicates that elemental fractionation during deep garnet-stable melting is insufficient to explain the observed REE composition (Supplementary Note 2). Even when considering the lowest realistic degree of melting for TTGs (10%[11]) and an eclogitic source assemblage (lowest $D_{La}/D_{Sm}$ and $D_{Eu}/D_{Gd}$; Supplementary Data 3), the melts that would be produced from a basaltic source with an unfractionated MORB-like REE composition (La/Sm ≈ Sm/Yb ≈ $Eu_N/Eu^*$ ≈ 1) would only have moderately elevated La/Sm (ca. 3), Sm/Yb (ca. 4) and $Eu_N/Eu^*$ (ca. 1.2). Similar $Eu_N/Eu^*$ would be obtained for all lithologies when considering non-modal fractional melting (see the "Methods" section). The fractionation of restitic phases during melting does not fully explain the observations either. The isolation of residual garnet from the melt may produce much higher La/Sm and Sm/Yb in the melt[27], and may likewise explain high Sr/Y[27], but does not account for high $Eu_N/Eu^*$. Fractionation of other restitic phases such as clinopyroxene or amphibole would explain neither high $Eu_N/Eu^*$ nor fractionated REE compositions ($D_{MREE} ≈ D_{HREE}$; Supplementary Data 3). Accumulation of peritectic

minerals and the assimilation of country rock prior to fractional crystallisation were explored as a possible cause for fractionated REE and high $Eu_N/Eu^*$, but tests likewise proved negative. Common peritectic minerals that could impact the REE budget of the melt (e.g., clinopyroxene, amphibole, garnet) have $D_{La} < D_{Sm} < D_{Yb}$ (Supplementary Data 3), meaning that their accumulation would lower, rather than raise, the La/Sm and Sm/Yb values of the melt. Plagioclase accumulation or assimilation could be proposed to at least explain the positive Eu anomalies. However, we calculate that over 40% modal plagioclase (MORB-sourced; ca. 0.5 ppm Eu, and $Eu_N/Eu^*$ of ca. 7.5) would need to be ingested for a TTG melt (ca. 0.7 ppm Eu) that initially had no anomaly for it to develop a $Eu_N/Eu^*$ of ca. 1.6. Even more extensive plagioclase ingestion would be needed if plagioclase was sourced from other TTGs or other granitoids. Besides the fact that the expected correlation between $Eu_N/Eu^*$ and CaO, $Na_2O$ or Sr is not observed (Fig. S2), such degrees of assimilation would produce a plagiogranite rather than TTGs. These tests show that the REE composition of primary TTGs cannot be explained through melt-induced element fractionation and assimilation alone; the source rocks must also have had fractionated REE compositions and positive Eu anomalies and these signatures were variably amplified in the melt depending on the degree of melting and fractionation of restitic phases.

Of all possible source lithologies that TTGs may derive from, only mafic plagioclase cumulates fit the requirements of having sufficiently high La/Sm and Sm/Yb, and $Eu_N/Eu^*$. Such rocks may represent primary gabbro(-norites) or hybridised rocks formed by the interaction between lower-crustal pyroxene-plagioclase granulites and picritic melts[34]. Melt modelling confirms that single-stage melting at different degrees of sources with compositions typical of gabbroic or gabbro-noritic rocks can explain all compositional characteristics of low-$F$ TTGs, without requiring additional processes such as multi-stage accumulation and hybridisation[30] (Supplementary Note 3). Such rocks also account for the presence of small amounts of water during TTG melting[8]. Although generally considered "anhydrous", gabbroic rocks on average contain 2.3 wt% water, likely hosted in fluid inclusions and hydrous phases such as amphibole (Supplementary Note 3). The

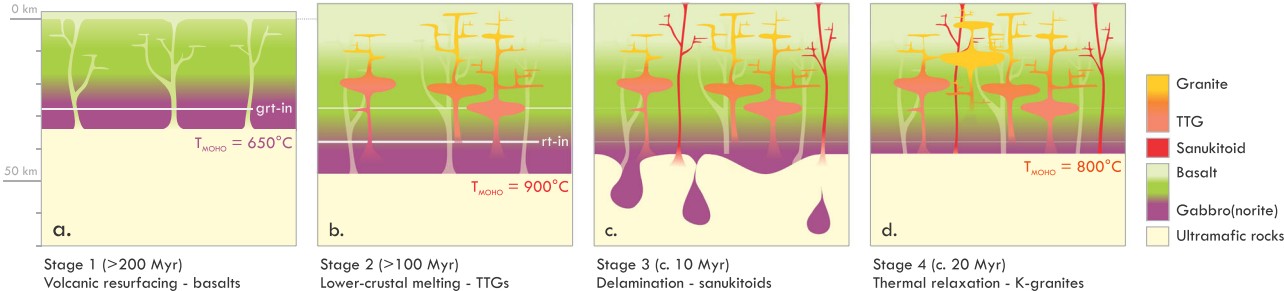

**Fig. 4 | Schematic representation of the process of TTG magmatism and the processes that followed. a** Development of the early crust and gradual burial of its mid- to lower-crust as a result of basaltic resurfacing (0 km reference = initial location of the base of the basaltic pile). **b** TTG magmatism caused by the melting of lower-crustal rocks that were heated beyond their solidus temperature due to the warming of the Moho, **c** Delamination of meta-gabbro restites and sanukitoid

magmatism caused by decompression melting of the metasomatised mantle that was welling up into areas of foundering, **d** thermal relaxation following removal of the crustal root causes low-degree melting of the middle crust, resulting in late-stage K-granite magmatism. Modal abundance of intrusions underestimated for figure clarity.

variably high Sr/Y in primary TTGs (Fig. 3c) would indicate a source that is largely, if not entirely, free of plagioclase. Consistent with this and the estimated depth of melting (≥35 km[27,39]), a particularly good fit between TTGs and model melts is obtained for gabbroic sources that comprise a hydrous eclogitic assemblage of garnet, clinopyroxene, amphibole and rutile. These findings are at odds with the interpretation of sources comprising hydrous (meta-)basalt[8,11,12,22,23,31] or peridotite[35]. The issue here is that the compositional features that would distinguish melts from a cumulate source from those derived from a basaltic or peridotitic source are subtle and are typically lost during fractional crystallisation and assimilation. An example of obscured cumulate sources is provided by TTGs from the Pilbara Craton[23]. These rocks are cogenetic with basalts in the region and were interpreted to derive from these on the basis of their major- and trace-element composition. The TTGs have only moderately elevated La/Sm and Sm/Yb and largely lack a positive Eu anomaly, which could indicate a basaltic source. Most of the rocks, however, prove to be significantly fractionated, with $F$ values of 15 and higher. There is only one sample that has a low $F$ value of ca. 2 and this is the only sample that still retains high La/Sm and Sm/Yb, as well as positive Eu anomaly. Such examples illustrate how magmatic differentiation may obscure primary TTG compositions and may bias compositions towards those that would suggest a non-cumulate source rock.

## Causes and consequences of TTG magmatism

The interpretations presented here confirm long-standing propositions of a petrogenetic link between plagioclase-bearing mafic cumulates and trondhjemites[45,46] and highlight the significance of field observations demonstrating the segregation of TTG melts from anatectic meta-gabbros[33]. The results furthermore stimulate revision of the consensus that TTGs derive from basaltic rocks[8,11,23,24] and set constraints on the petrogenetic setting in which TTGs were made. The melting of mafic plagioclase cumulates cannot be readily reconciled with a subduction setting. Such rocks—specifically meta-gabbros—are present in the subducting lower crust, but these are not involved in slab melting. The latter process is restricted to the slab-mantle interface, where upper-crustal lithologies with low solidus temperatures, such as hydrous sediments and hydrothermally altered basalts, are exposed to the hot mantle wedge. If at all reactive, the gabbros of the slab lower crust may undergo fluid-assisted metamorphism and deformation[47–49]. They nevertheless would not melt, because—even at the high subduction geotherm expected for hot Archaean subduction zones (10–12 °C/km[8])—slab Moho temperatures remain well below solidus temperatures[50]. The melting of mafic cumulates to produce TTGs thus must be sought among the lower proto-continental crust. Here, we consider the petrogenesis, metamorphism and melting of the TTG sources as part of a causal chain that was set in motion by the

volcanic resurfacing of basaltic material over mafic proto-continents[3,21].

For the mantle potential temperatures that likely occurred during the Archaean (>1600 °C[51]), the proto-continental crust is predicted to have been 25–35 km thick[51]. Crustal lids on the thick side of this range likely formed over domains of active mantle upwelling[52], the chemical signature of which is recorded in the voluminous high-Ti basalts and komatiites observed in Archaean greenstone belts[53]. This proto-continental crust was poorly differentiated and mafic on average, and contained a thick basal layer made of gabbros and other mafic cumulates (Fig. 4a). These rocks exhibit mantle-like $\delta^{18}O$ values[54] and were buried due to volcanic resurfacing[3]. Taking the example of a proto-crust formed over a plume-like upwelling (ca. 35 km thick), we calculate that the volcanic resurfacing of a basalt pile of a realistic thickness (ca. 10 km[3,8,53]) would depress the Moho of such crust beyond 40 km depth, allowing it to reach $P$–$T$ conditions at which rutile is stable[43]. Assuming a general Archaean geotherm of ca. 20 °C/km[55], this burial would heat the Moho from an initial 700 to ca. 900 °C, which exceeds solidus temperatures of hydrous gabbro[56]. Any gabbroic domains of the lower crust that were not completely anhydrous would melt to a degree that is likely controlled by water content. This melting would occur at garnet- and rutile-stable conditions and would produce primary TTGs with high (Ti$_N$/Ti*)/Nb and Eu$_N$/Eu*, and variably fractionated REE composition (Fig. 4b). We note here that, although crustal drips may likely have occurred at this stage[17,21,24], this process is not necessary to explain deep-crustal melting at given conditions. The melt would inherit the isotopically light O of the source, thus explaining the prevalence of low $\delta^{18}O$ values in TTGs[54]. Hafnium and Sr isotope data indicate that the turn-around time from making the gabbroic source and starting volcanic resurfacing to TTG magmatism was 0.3–0.5 Gyr on average[57,58], which is consistent with the duration of basaltic magmatism in Archaean greenstone belts[53]. Episodes of TTG magmatism typically last well over 100 Myr[15]. Increasing Nb/Ta during such episodes was suggested to indicate a deepening of the source rock[20] into rutile-stable conditions. Low-$F$ TTGs nevertheless cover a large range in Nb/Ta (5–45[44]) and all of these indicate the presence of residual rutile. Increasing Nb/Ta in melts from rutile-bearing sources could instead relate to the steady depletion of the source. The $D_{Nb}$ for rutile in hydrous mafic systems is lower than $D_{Ta}$ at $P$–$T$ conditions relevant to TTG melting[59]. The counter-correlation between Nb/Ta and $\varepsilon_{Hf}$ in Isua TTGs thus may record the progressive depletion and formation of refractory rutile in the mafic source as it aged.

Extensive melt extraction from the gabbroic crustal roots does not go without consequence and can be considered as the prerequisite for the sanukitoid and K-rich granite magmatism that typically follows TTG magmatic episodes[13]. Melt extraction from the mafic lower crust leaves dense eclogitic residues that will founder into the

mantle[3,21,34,60,61] (Fig. 4c), which may explain the presence of eclogites in the mantle absent of subduction[3,34]. Supporting the link between these eclogites and TTGs is the observation that such eclogites, like the cumulate source rocks of TTGs, typically exhibit positive Eu anomalies[62]. Delamination has been proposed as a driver for long-lived or renewed TTG magmatism[17,21], or as a means of transporting TTG sources to great depth and producing sanukitoids by hybridising TTG melts from such depths with asthenosphere-derived liquids[8,24]. Neither of these mechanisms, however, occurs in modern examples of delamination. In modern analogues, the delamination of refractory crustal roots is a relatively short-lived process that is occasionally followed by a brief (0–10 Myr) episode of localised high-Mg andesitic or high-K basaltic magmatism. The lavas produced by such magmatism are sourced in the upwelling mantle contaminated with recycled crustal material[60,61,63,64]. Rare melt inclusions found in pyroxene phenocrysts within such unusual lavas in the Pamir[65], which erupted upon the foundering of the lower crust[63,64], provide a sample of such melts prior to their crystal fractionation. These inclusions exhibit an unusual composition, characterised by extreme enrichment in Ba, Th, U, K and REE, but also by high Cr concentrations. These features, as well as the relatively short magmatic duration, are the hallmarks of Archaean sanukitoid magmatism[6,35,66]. Where produced and preserved, sanukitoids thus may serve as rare and valuable magmatic markers of crustal delamination from the base of Archaean cratons (Fig. 4c). Although sanukitoids are mantle-derived melts[35,66], their Nb concentrations and $Ti_N/Ti^*$ values resemble those of Archaean granitoids (Supplementary Note 1). The incompatible-element composition of sanukitoids thus may be strongly controlled by the crustal fragments that foundered into the lithospheric mantle before and during terminal lower-crustal delamination. Delamination leaves a shallower, warmer Moho and a perturbed crustal geotherm, which—upon thermal relaxation—would cause the still-fertile middle crust to melt (Fig. 4d). We propose that this is the cause for young "granite blooms" in TTG terranes—episodes of high-K, low-Na granite magmatism that occur in the wake of TTG magmatism. Unlike the sources of TTGs, which are mantle-derived, the source rocks of the high-K granites would include buried sediments and other surface-altered materials, which have high $\delta^{18}O$ and contain old detritus. The transition in melting processes in the wake of delamination thus can explain the increase in $\delta^{18}O$ values and concomitant loss of juvenile Hf isotope compositions in late K-rich granites within TTG terranes[54,67]. If Archaean TTGs indeed formed by the melting of the gabbroic lower crust, then K-rich granitoid magmatism and the development of "modern" andesitic continental crust from mafic proto-crust likely represent natural consequences of the magmatic differentiation of the crust. Together, these processes comprise a fully endogenic mechanism that can explain sustained crustal growth and cratonisation during the Archaean, without requiring external forcings, such as meteorite impacts or the start of plate tectonics.

## Methods

The effects of partial melting and fractional crystallisation are quantified through Eqs. (1) and (2), respectively. In these equations, $C_x^l$ represents the concentration of element $x$ in the (remaining) liquid l, $C_{x,i}^l$ is the concentration of element $x$ in the liquid before fractional crystallisation, $C_x^s$ is the concentration in the solids within the source, $D_x$ is the distribution coefficient of the element $x$ in a given phase (or bulk $D$ weighted to the mineral modal abundance to prevent magnitude bias), and $f$ is the degree of melting (Eq. (1)) or the proportion of melt remaining (Eq. (2)). Partitioning coefficients for Ti, Nb, Zr, and Gd for rutile and ilmenite in basaltic systems, which are used to constrain the residual titaniferous phase in the source (Fig. 1b) are provided in Supplementary Data 1. Partition coefficients for the same elements in andesitic-dacitic systems, which are used to quantify the effects of fractional crystallisation in TTGs (Fig. 1c), are provided in Supplementary Data 2. Additional partition coefficients for mafic systems

used to model source compositions are provided in Supplementary Data 3. The effects of non-modality on the REE composition of TTG melts were investigated through non-modal melt modelling using Eq. (3), where $D_0$ is the bulk $D$ for element $x$ at the onset of melting and $P$ is the bulk distribution coefficient for that element in the melting assemblage, both weighted by mineral modal abundance in said melting assemblage. These calculations were done assuming the melting assemblage comprised 75% plagioclase (garnet-bearing amphibolite and granulite), or 75% amphibole (eclogite), both complemented with clinopyroxene. The degree of fractional crystallisation in TTGs and other granitoids is quantified through the fractionation factor $F$, where $x$ represents a given TTG and the scalar (10) approximates unfractionated TTGs that have not undergone fractional crystallisation. The fractionation factor $F$ is calculated following Eq. (4) in which $Nb_x$ and $(Ti_N/Ti^*)_x$ are the Nb concentration and Ti anomaly ratio in a given TTG, and $Nb_{GEM}$ and $(Ti_N/Ti^*)_{GEM}$ are the Nb concentration (8 ppm) and Ti anomaly ratio (1) of the Granitoid End Member reference.

$$C_x^l = C_{x,i}^l f^{(D_x - 1)} \tag{1}$$

$$C_x^l = \frac{C_x^s}{(1 - D_x)f + D_x} \tag{2}$$

$$C_x^l = \frac{C_x^s}{D_0}\left(1 - \frac{Pf}{D_0}\right)^{(P^{-1} - 1)} \tag{3}$$

$$F = 10 - \frac{Nb_x - Nb_{GEM}}{\frac{Ti_N}{Ti^*}_x - \frac{Ti_N}{Ti^*}_{GEM}} \tag{4}$$

## Data availability

Unless noted otherwise, the data used in this study were obtained, and are available from, the Geochemistry of Rocks of the Oceans and Continents (GEOROC) geochemical data repository hosted by the Georg-August-Universität, Göttingen (https://georoc.eu/). Use of these data is permitted under the regulations of the Creative Commons Attribution-ShareAlike 4.0 International (CC BY-SA 4.0). Compilations of data in spreadsheet format are also available on request through the authors.

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

## Acknowledgements
Helpful discussions with A. Maltese, J. Halla, A. Petersson, S. Weatherley, S. Caton and E. Kooijman, and constructive reviews by B.S. Kamber and an anonymous reviewer enabled significant improvements to the manuscript. Financial support was provided by the National Science and Engineering Research Council of Canada (Discovery Grant RGPIN-2020-04692 and Accelerator Grant RGPAS-2020-00069 to M.A.S.) and the University of British Columbia (International Doctoral Fellowship to K.A.M.).

## Author contributions
M.A.S. developed concepts, performed data collection, evaluation and synthesis, and prepared the first draft text. K.A.M. and J.G. contributed to concept development and revised and edited the text.

## Competing interests
The authors declare no competing interests.
