## [Peer Review File · Nature Communications]

REVIEWER COMMENTS

Reviewer #1 (Remarks to the Author):

Here comes my review of the manuscript "Archean continental crust formed from mafic cumulates" by Dr. Smit and colleagues

The thought of using high field-strength elements to quantify the cumulative effect of crystal fractional crystallization is quite interesting and novel. The topic is of broad international interest and suitable for the Nature Communications. However, I am not totally convinced that the interpretation here is the only way to interpret the data, since some important geological processes are neglected. I am afraid that this manuscript cannot be accepted as presented. My main concerns and objection are list below, along with minor comments.

MAIN CONCERNS

(1) The authors try to quantify the effect of fractional crystallization from partial melting processes. It seems that only fractional crystallization would change the composition of the initial melts. This is oversimplified. Mineral accumulation, like plagioclase, has been demonstrated to be able to produce high pressure like TTGs (Laurent et al., 2019, Nature Geoscience; Kendrick et al., 2022, Geology). The authors should not only consider removal of minerals or mineral assemblage. The assimilation of the country rocks and entrainment of peritectic minerals like garnet, rutile would also change the initial melt compositions.

(2) The selection of the partition coefficient is odd. The authors list a lot of partition coefficients from the literature, and use the mean value of these partition coefficients. However, the selection of the partition coefficients should be suitable to the system. Many of these partition coefficients are not used for high silica melt.

(3) When consider basaltic rock as the source for TTGs, the authors always forget the importance of plagioclase. Throughout the manuscript, the authors state that the high pressure TTG can be produced by garnet-stable melting of basaltic crust. This is not exact. High pressure TTGs should be produced at garnet-bearing and plagioclase-absent field. Since plagioclase strongly incorporate Sr and garnet incorporate Y, high pressure TTGs have high Sr/Y. The state that Sr/Y is poor indicator of melting depth is wrong, especially the authors could quantify the fractional crystallization.

(4) The authors state that basaltic source is ascribed to subducting mafic crust, and gabbroic source cannot be reconciled with a subduction setting. The difference between basalt and gabbro is that basalts are extrusive igneous rock and gabbros are intrusive rock. The compositions of basalt are equivalent to gabbro. Subduction is not necessary for partial melting of basalt. Archaean drip is sufficient for basaltic crust sinking to partial melting zone. Some important papers have been missed, such as Johnson et al., 2014, Nature Geoscience; Smithies et al., 2021, Nature.

(5) The positive Eu/Eu^* are observed in unfractionated, low-F TTGs. The authors state that this must be characteristics of the source rocks. This is not the only way to produce positive Eu/Eu^* TTGs. Plagioclase accumulation could also induce positive Eu/Eu^* TTGs (Kendrick et al., 2022, Geology).

Minor comments

Line 1. Archean. As Nature Communication is published in the UK, it should be spelled 'Archaean' in UK spelling.

Line 35. Even Hadean. The Acasta gneiss.

Line 38. The mafic source should be not only tholeiitic, but enriched in LILEs.

Line 41. Garnet-bearing and plagioclase-absent field.

Line 74. Why two 'lower' here?

Line 77. Not all the partition coefficients list in the table is suitable for high silica melt.

Line 97. Nope. The stability field of titanite will depend on the calcium content in the proto composition. High Ca basalt will produce titanite in this condition (see Huang et al., 2020, CTMP).

Line 101. The authors should not use capital letters in Figure. It should be Fig.1a.

Line 114. Nope. The authors should also consider mineral accumulation here, such as plagioclase.

Line 161-165. The reason for the doubt is the selection of partition coefficient. Some of the partition coefficients list in Table S2 are not suitable for granitic melts. If you use the partition coefficient from Bedard (2006, GCA), Yb is compatible in amphibole.

Line 168. Do you mean EuN/Eu^* ?

Line 168. In Supplementary Fig. 2, the caption labels $F > 5$.

Line 171. As mentioned above, Sr/Y is not poor indicator of melting depth.

Line 198. Not necessary. Plagioclase accumulation can induce positive Eu/Eu^* .

Line 213. How can a gabbro buried at 35 km get 2.3wt% of water?

Line 222. This is not the fact in these papers. These papers never link basalts to subduction setting.

Line 228. Reference need to be cited for this geotherm. This is important.

Line 242. Also. Reference is necessary for the geotherm.

Reviewer #2 (Remarks to the Author):

This manuscript investigates Ti/Ti* and Ti/Nb systematics of granitoids in the context of the TTG, whose petrologic origin lies at the heart of the debate regarding Archaean continental crust formation. This is a topic of wide interest and makes the submission suitable for the journal. Whereas I find myself agreeing with the overall conclusion, and like the novelty of the $\log(\text{TiN}/\text{Ti}^*)$ vs. $\log(\text{Nb})$ diagram, I cannot recommend acceptance of the m/s in its present form and would suggest that any potential resubmission would have to be reviewed ab initio. I arrived at this impression for three general reasons.

Firstly, the m/s misses reference to some very relevant recent publications on the use of phase equilibrium modelling for the origin of Archaean TTGs and related rocks. Important papers not (sufficiently) considered are:

Palin, R.M., White, R.W., Green, E.C., Diener, J.F., Powell, R. and Holland, T.J., 2016. High-grade metamorphism and partial melting of basic and intermediate rocks. *Journal of Metamorphic Geology*, 34(9), pp.871-892.

Kendrick, J. and Yakymchuk, C., 2020. Garnet fractionation, progressive melt loss and bulk composition variations in anatectic metabasites: Complications for interpreting the geodynamic significance of TTGs. *Geoscience Frontiers*, 11(3), pp.745-763.

Hernández-Montenegro, J.D., Palin, R.M., Zuluaga, C.A. and Hernández-Urbe, D., 2021. Archean continental crust formed by magma hybridization and voluminous partial melting. *Scientific Reports*, 11(1), p.5263.

Emo, R.B. and Kamber, B.S., 2022. Linking granulites, intraplate magmatism, and bi-mineralic eclogites with a thermodynamic-petrological model of melt-solid interaction at the base of anorogenic lower continental crust. *Earth and Planetary Science Letters*, 594, p.117742.

Kendrick, J., Duguet, M. and Yakymchuk, C., 2022. Diversification of Archean tonalite-trondhjemite-granodiorite suites in a mushy middle crust. *Geology*, 50(1), pp.76-80.

Triantafyllou, A., Ducea, M.N., Jepsen, G., Hernández-Montenegro, J.D., Bisch, A. and Ganne, J., 2022. Europium anomalies in detrital zircons record major transitions in Earth geodynamics at 2.5 Ga and 0.9 Ga. *Geology*.

I would normally side with the authors in giving credit to early, original work but in this instance, there is so much progress being made with phase equilibrium melting above the solidus of metabasites that it is impossible to evaluate the uniqueness and novelty of the material presented in the m/s in the context of these new papers. Only the earliest of these (Palin et al., 2016) is mentioned in the m/s but not in the context of the envisaged phase assemblages to explain the $\log(\text{TiN}/\text{Ti}^*)$ vs. $\log(\text{Nb})$ systematics. Obvious omissions are comparison with the outputs of the Kendrick and Yakymchuk (2020) model and the basaltic underplate-gabbro residue hybridisation model of Emo and Kamber (2022), which also modelled eclogite delamination. I suggest that a new submission should frame the TTG debate around these latest insights and then advance the new $\log(\text{TiN}/\text{Ti}^*)$ vs. $\log(\text{Nb})$ data in that context.

Secondly, I am unclear why the authors did not consider the inclusion of Ta, e.g., via. Nb/Ta? I would have thought that there would be sufficient high-quality data for TTG and non-TTG granitoids to test whether Nb/Ta instead of [Nb] would yield even more discrimination power? At the very least, more context needs to be given to Nb/Ta, as rutile and ilmenite have different Ds not just for Nb but for Ta also, i.e., the foundation of the Hoffmann et al. (2011) model, which is cited as [19] but not really discussed in a meaningful way, i.e. as a testable hypothesis.

Thirdly, I was surprised to see the authors opt to calculate Ti^* relative to Zr and Eu. I can understand the inclusion of Zr instead of a MREE but am unclear why Eu was preferred over Gd, as this choice now 'marries' the effects of Ti-phases and plagioclase. I think a revised submission would need to provide a diagram equivalent to Figure 1 with TiN/Ti^* calculated with Ti^* as $\sqrt{\text{GdN} \times \text{ZrN}}$ so that readers (and reviewers) can scrutinise the authors' claim that their preferred Ti^* is indeed superior.

I hope the authors will find my suggestions useful to produce a more compelling case.

Submission revised manuscript NCOMMS-23-25163

Sept 29th, 2023

REVIEWER

Comment number

Original comment or suggestion.

Response to comment.

→ Summary of concrete changes made with line numbers.

REVIEWER 1

Main comment 1.1

"The authors try to quantify the effect of fractional crystallization from partial melting processes. It seems that only fractional crystallization would change the composition of the initial melts. This is oversimplified. Mineral accumulation, like plagioclase, has been demonstrated to be able to produce high pressure like TTGs (Laurent et al., 2019, Nature Geoscience; Kendrick et al., 2022, Geology). The authors should not only consider removal of minerals or mineral assemblage. The assimilation of the country rocks and entrainment of peritectic minerals like garnet, rutile would also change the initial melt compositions. "

This is a very valuable comment and we admit to not have given this significant attention. We now consider the effects of the assimilation of country rocks and peritectic minerals as well (and have added said references). We do this across the board; assimilation is now considered in the interpretation of Nb-Ti systematics, which is used to constrain the stability of titaniferous phases in the TTG source, and we also consider assimilation of all major silicates, including plagioclase, in the interpretation of the REE composition of primary TTG melts. We performed assimilation calculations to test the significance of these processes in each of these occasions. Although the results of these calculations demonstrate that the cause of the observed signatures must be sought elsewhere, we consider including assimilation considerations a very valuable addition that made the manuscript much more comprehensive and robust.

→ New text was added in lines 106-116, 152-154, 220-233, and 254-257.

Main comment 1.2

"The selection of the partition coefficient is odd. The authors list a lot of partition coefficients from the literature, and use the mean value of these partition coefficients. However, the selection of the partition coefficients should be suitable to the system. Many of these partition coefficients are not used for high silica melt."

We recognize that our choice and use of the partition coefficients may not have been clearly explained. Two sets of partition coefficients were used: 1) one set for basaltic systems taken to represent melting of the mafic source and 2) another set for andesitic-dacitic (=TTG-like) systems, which were used to investigate the chemical effects of fractional crystallization on Nb-Ti-Zr-Eu (now Nb-Ti-Zr-Gd, see *Main Comment 2.3*) in TTG melts. The appropriate partition coefficients were used in each case, but this was not concretely indicated and the appropriate Nb-Ti-Zr-Gd partition coefficients for silicate minerals in andesitic-dacitic systems were missing. This is now rectified. The effects of silicate fractional crystallization on various element ratios in TTGs are only considered qualitatively, because, although absolute D values may differ between intermediate and felsic systems, differences in D values between minerals typically remain the same. Our qualitative assessment thus allows evaluation of changes in element ratios due to fractional crystallization of amphibole and plagioclase independent of the exact SiO_2 concentration of the melt. The text has been changed to reflect that.

→ New text was added in lines 100-106, 108-112, and 116-120. The use of partition coefficients for different purposes and systems is now explicitly noted in the Methods (372-378). A new Supplementary Table was made (Table S2) and the captions of all Supplementary Tables were made clearer.

Main comment 1.3

"When consider basaltic rock as the source for TTGs, the authors always forget the importance of plagioclase. Throughout the manuscript, the authors state that the high pressure TTG can be produced by garnet-stable melting of basaltic crust. This is not exact. High pressure TTGs should be produced at garnet-bearing and plagioclase-absent field. Since plagioclase strongly incorporate Sr and garnet incorporate Y, high pressure TTGs have high Sr/Y. The state that Sr/Y is poor indicator of melting depth is wrong, especially the authors could quantify the fractional crystallization."

We agree with the reviewer that garnet and plagioclase control Sr/Y in TTGs. In mafic source rocks undergoing migmatization, garnet will provide a residual aluminous phase formed by the dehydration of amphibole – a reaction that consumes plagioclase. We, however, also note that aspects other than pressure may control Sr/Y in TTGs (e.g., Kendrick & Yakymchuk, 2020, as suggested by Reviewer 2). This conclusion is supported by the variability in Sr/Y among low- F TTGs – all of which formed at rutile-stable (and thus garnet-stable) depth. We agree that our original statement on this lacked nuance. The text was revised and new context provided in this regard.

→ New text was added in lines 52-56, 195-200, and 249-253.

Main comment 1.4a

"The authors state that basaltic source is ascribed to subducting mafic crust, and gabbroic source cannot be reconciled with a subduction setting. The difference between basalt and gabbro is that basalts are extrusive igneous rock and gabbros are intrusive rock. The compositions of basalt are equivalent to gabbro. Subduction is not necessary for partial melting of basalt. [...]"

Basaltic and gabbroic rocks are indeed compositionally similar, but there are subtle differences, which show the complementary nature of these rocks as well. One of these is a positive Eu anomaly, which is absent in MORB and other common basalt types, yet is characteristic of gabbros and other plagioclase-cumulate intrusive rocks. Having identified the latter as TTG source rocks actually does have significant implications for the setting of melting. For instance, gabbros do not participate in slab melting inside subduction zones. Although this is fairly common knowledge in the subduction-zone community, we agree that this could have been explained in a bit more detail. We now do so in our more comprehensive discussion of this aspect.

→ New text was added in lines 275-285.

Main comment 1.4b

"[...] Archean drip is sufficient for basaltic crust sinking to partial melting zone. Some important papers have been missed, such as Johnson et al., 2014, Nature Geoscience; Smithies et al., 2021, Nature."

We agree that crustal drips may explain such sinking, but we would also add that this process is not necessary to bury rocks down to the conditions indicated for TTG melting. We demonstrate this quantitatively in the discussion (lines 288-298), which is based on reliable *P-T* estimates for TTG melting. We do not rule out that drips occurred and played some role in TTG magmatism (and do make a note of this now). However, given that the explained burial process already provides the same prerequisites for melting without requiring drips, we consider this process as governing in global Archean TTG magmatism. The suggested references indeed were not included and are now incorporated in various places. The $\delta^{18}\text{O}$ study (Smithies et al., 2021), and those like it (Kemp et al., 2007), provided a particularly valuable addition, as it is consistent with the proposed transition in melting process going from TTG to potassic granite magmatism.

→ References and their content are included in lines 48-52, 285-287, 304-306, 320-322, 327-330 (Johnson et al., 2014), and lines 292-295, 306-308 and 351-356 (Smithies et al., 2021).

Main comment 1.5

"The positive Eu/Eu are observed in unfractionated, low-F TTGs. The authors state that this must be characteristics of the source rocks. This is not the only way to produce positive Eu/Eu* TTGs. Plagioclase accumulation could also induce positive Eu/Eu* TTGs (Kendrick et al., 2022, Geology)."*

As *per* our explanation regarding Main comment 1.1, we now also discuss this process and have included the reference as part of this.

→ New text regarding assimilation was added in lines 106-116, 152-154, 220-233, and 254-257.

Minor comment 1.1

"Line 1. Archean. As Nature Communication is published in the UK, it should be spelled 'Archaean' in UK spelling."

Thank you for this note. The reviewer is correct and changes have been made accordingly.

→ The change has been applied throughout the manuscript.

Minor comment 1.2

"Line 35. Even Hadean. The Acasta gneiss."

The reviewer is correct and the suggested change was made.

→ The change was made to the text in lines 34-36.

Minor comment 1.3

"Line 38. The mafic source should be not only tholeiitic, but enriched in LILEs."

The reviewer is correct and the suggested change was made.
→ The change was made to the text in lines 38-42.

Minor comment 1.4

"Line 41. Garnet-bearing and plagioclase-absent field."

The reviewer is correct and the suggested change was made.
→ The change was made to the text in lines 44-45.

Minor comment 1.5

"Line 74. Why two 'lower' here?"

This was an error. The duplicate "lower" was removed.
→ The change was made to the text in lines 92-94.

Minor comment 1.6

"Not all the partition coefficients list in the table is suitable for high silica melt."

As *per* our explanation regarding Main comment 1.2, the partition coefficients indeed were not all suitable for high-Si melt. Part of these are for basaltic systems (melting calculations) and part of these are for TTG systems (fractional crystallization calculations). This is now explicitly stated and explained in the newly added text.
→ New text was added in lines 100-106, 108-112, and 116-120. The use of partition coefficients for different purposes and systems is now explicitly noted in the Methods (372-378). A new Supplementary Table was made (Table S2) and the captions of all Supplementary Tables were made clearer.

Minor comment 1.8

"Line 97. Nope [sic]. The stability field of titanite will depend on the calcium content in the proto composition. High Ca basalt will produce titanite in this condition (see Huang et al., 2020, CTMP)."

A great suggestion. We did not first consider this paper, but it provides valuable insights. Titanite is indeed modelled and observed to be stable in Ca-rich basaltic rocks. However, consistent with the findings from the studies that we did include (Liou et al., 1998; John et al., 2011), both the thermodynamic modelling and the experiments done by Huang et al. (2020) show that titanite is lost at temperatures above 750°C. Addition of this information provides a more comprehensive argument to focus on ilmenite and rutile as possible titaniferous residual phases.
→ The reference and new text explaining the findings were added in lines 95-99.

Minor comment 1.9

"Line 101. The authors should not use capital letters in Figure. It should be Fig.1 a."

We were unaware and are grateful for this note.
→ The suggested changes were made throughout the manuscript, as well as in the supplementary files.

Minor comment 1.10

"Line 114. Nope [sic]. The authors should also consider mineral accumulation here, such as plagioclase."

As *per* our responses to Main comments 1.1 and 1.5, we now also consider the effects of mineral accumulation, both with regards to its effects on Nb-(Ti_N/Ti*) and REE.

→ New text regarding assimilation was added in lines 106-116, 152-154, 220-233, and 254-257.

Minor comment 1.11

"Line 161-165. The reason for the doubt is the selection of partition coefficient. Some of the partition coefficients list in Table S2 are not suitable for granitic melts. If you use the partition coefficient from Bedard (2006, GCA), Yb is compatible in amphibole."

As *per* our response to Main comment 1.2 and Minor comment 1.6, we have used partition coefficients that are suitable for the given system (basaltic for melting of the source and andesitic-dacitic for fractional crystallization in TTG melts). The Bédard (2006) partition coefficients apply to intermediate compositions (c. 60 wt% SiO₂) and thus are not suitable for calculating element partitioning during partial melting of basaltic sources.

→ New text was added in lines 100-106, 108-112, and 116-120. The use of partition coefficients for different purposes and systems is now explicitly noted in the Methods (372-379). A new Supplementary Table was made (Table S2) and the captions of all Supplementary Tables were made clearer.

Minor comment 1.12

"Line 168. Do you mean Eu_N/Eu?"*

Correct. The "*" was indeed missing there. There error is corrected.

→ The change was made to the text in lines 203-207.

Minor comment 1.13

"Line 168. In Supplementary Fig. 2, the caption labels F>5."

Correct. The "<" was indeed an error. It should have read ">". In retrospect, we decided to switch to "primary TTG" in the discussion part of the manuscript, because this is easier to read than "low-*F* TTG". There error is corrected.

→ The change was made to the caption of Fig. S2.

Minor comment 1.14

"Line 171. As mentioned above, Sr/Y is not poor indicator of melting depth."

As *per* our response to Main comment 1.3, we have removed this text and now provide a more nuanced and accurate perspective on Sr/Y and its meaning. We do note that opinions regarding this aspect differ and there is evidence to the contrary (see our addressing of Main comment 1.3).

→ New text regarding Sr/Y and its interpretation was added in lines 52-56, 195-200, and 249-253.

Minor comment 1.15

"Line 198. Not necessary. Plagioclase accumulation can induce positive Eu/Eu."*

As *per* our responses to Main comments 1.1 and 1.5, and Minor comment 1.10, we now also consider the effects of mineral accumulation, and the effects of plagioclase on Eu_N/Eu* in particular.

→ New text regarding assimilation was added in lines 106-116, 152-154, 220-233 (specifically about Eu_N/Eu*), and 254-257.

Minor comment 1.16

"Line 213. How can a gabbro buried at 35 km get 2.3wt% of water?"

This is an interesting question and possibly a topic of a separate study. For this study, the observation alone is needed. Gabbros and gabbronorites are often considered "anhydrous", even though many of these contain some hydrous phase – typically amphibole, but sometimes also phlogopite and in Fe-Ti gabbros epidote. It is likely that

these phases – and amphibole in particular – participate in the melting process, thus likely introducing the small amount of H₂O that experiments indicate to be present during TTG melting. Although this was already noted in the Supplementary Text, we do agree that this is an important aspect that could do with a bit of clarification. We therefore added new text in both the manuscript and the Supplementary Text 4 to clarify the hydrous component that is present in gabbros, regardless of their petrogenetic setting.

→ New text was added in lines 246-249.

Minor comment 1.17

"Line 222. This is not the fact in these papers. These papers never link basalts to subduction setting."

We agree that, of these papers, only Foley et al. (2002) implicitly explains their observations in the framework of "subduction-zone environments". Rapp et al. (1991) notes this as a setting that the Earth evolves into having and that TTGs may be produced in such settings. The note did not seem of vital importance, so we have removed the text and references here for the sake of clarity and accuracy.

→ The text and references were omitted.

Minor comment 1.18

"Line 228. Reference need to be cited for this geotherm. This is important."

The geotherm was obtained from Moyen & Martin (2012), who propose such a geotherm for hypothetical Archean subduction.

→ The references was added in the text in lines 284-286.

Minor comment 1.19

"Line 228. Reference need to be cited for this geotherm. This is important."

The geotherm was proposed as a general average for Archean cratons by Sun et al. (2019).

→ The references was added in the text in lines 298-300.

REVIEWER 2

Main comment 2.1

"Firstly, the m/s misses reference to some very relevant recent publications on the use of phase equilibrium modelling for the origin of Archaean TTGs and related rocks. Important papers not (sufficiently) considered are: [...]. I would normally side with the authors in giving credit to early, original work but in this instance, there is so much progress being made with phase equilibrium melting above the solidus of metabasites that it is impossible to evaluate the uniqueness and novelty of the material presented in the m/s in the context of these new papers. Only the earliest of these (Palin et al., 2016) is mentioned in the m/s but not in the context of the envisaged phase assemblages to explain the log(TiN/Ti) vs. log(Nb) systematics. Obvious omissions are comparison with the outputs of the Kendrick and Yakymchuk (2020) model and the basaltic underplate-gabbro residue hybridisation model of Emo and Kamber (2022), which also modelled eclogite delamination. I suggest that a new submission should frame the TTG debate around these latest insights and then advance the new log(TiN/Ti*) vs. log(Nb) data in that context."*

This is a very valuable comment and we admit to not have given these important papers significant attention. We incorporated all of these papers, either by including them, or by using their content more effectively. The addition of this content allowed for a much more comprehensive and up-to-date framework, and provided a few valuable additions that are consistent with our findings and thus illustrate internal consistency. Changes made regarding each of the mentioned papers are listed below. Together, these changes allow us to better convey the uniqueness, novelty and importance of our study.

- Palin, R.M., White, R.W., Green, E.C., Diener, J.F., Powell, R. and Holland, T.J., 2016. High-grade metamorphism and partial melting of basic and intermediate rocks. *Journal of Metamorphic Geology*, 34(9), pp.871-892.
 - This paper provides many insight into the phase assemblages that may be stable in basaltic sources. We used the paper specifically for its evidence that: 1) TTGs may indeed be formed by MORB melting (lines 63-65), 2) titanite is lost from potential TTG sources during melting at temperatures in excess of 750°C (lines 96-99), and 3) rutile is stable in basaltic sources at 1.4 GPa or higher (lines 197-199). The paper was also included as an additional reference in favour of sources being basaltic (lines 253-254).
- Kendrick, J. and Yakymchuk, C., 2020. Garnet fractionation, progressive melt loss and bulk composition variations in anatectic metabasites: Complications for interpreting the geodynamic significance of TTGs. *Geoscience Frontiers*, 11(3), pp.745-763.
 - This paper was used for its evidence that fractionated REE and high Sr/Y, which are conventionally used to indicate high melting pressure, may already form at c. 1.4 GPa (lines 54-56). This is an essential addition that demonstrates that 1) melting depth of TTGs has likely been overestimated in the past and 2) the depths proposed in our model (c. 40 km) could realistically produce melts with fractionated REE. The paper was additionally used as evidence consistent with our new insights that REE ratios are not necessarily depth-indicative (lines 195-200), for its insights into the effects of garnet fractionation during melting (lines 216-218) and for its melting-depth estimates (lines 251-253).
- Hernández-Montenegro, J.D., Palin, R.M., Zuluaga, C.A. and Hernández-Urbe, D., 2021. Archean continental crust formed by magma hybridization and voluminous partial melting. *Scientific Reports*, 11(1), p.5263.
 - This paper was used for showing that melt hybridization, rather than single-stage melting and crystallization, may explain TTG compositions (lines 59-62). We also mention the reference again later when we explain that our relatively simple model can explain TTG compositions without requiring such complicated processes (lines 242-246).
- Emo, R.B. and Kamber, B.S., 2022. Linking granulites, intraplate magmatism, and bi-mineralic eclogites with a thermodynamic-petrological model of melt-solid interaction at the base of anorogenic lower continental crust. *Earth and Planetary Science Letters*, 594, p.117742.
 - This paper provided very useful insights and was used for its explanation that TTG sources may be refractory domains of the lower crust that were hybridized through interacting with picritic basalts (lines 69-72). This process may well explain the TTG source, as indicated in lines 239-242. The paper also provides very valuable insights into the processes and effects of delamination, and thus was added as a resource in various parts of the specific section of the discussion that handles that topic (lines 322-325).
- Kendrick, J., Duguet, M. and Yakymchuk, C., 2022. Diversification of Archean tonalite-trondhjemite-granodiorite suites in a mushy middle crust. *Geology*, 50(1), pp.76-80.
 - This paper was used for its insights into the role of plagioclase accumulation on TTG compositions (lines 59-62, 74-77, 172-174, 181-184). The arguments that lead us to prefer our interpretation are discussed as well (lines 144-147).
- Triantafyllou, A., Ducea, M.N., Jepson, G., Hernández-Montenegro, J.D., Bisch, A. and Ganne, J., 2022. Europium anomalies in detrital zircons record major transitions in Earth geodynamics at 2.5 Ga and 0.9 Ga. *Geology*.
 - This paper was used for its evidence that Eu_N/Eu^* is a poor depth indicator (lines 56-59). To provide the proper perspective, we also include a milestone paper (Moyen & Stevens, 2006) that explicitly considers Eu_N/Eu^* as depth indicator (lines 38-42). We agree with Triantafyllou et al. (2022) that source rock composition has a strong control on Eu_N/Eu^* and explain this, as well as related aspects, in the new text under header "Mafic plagioclase-cumulate source rocks" (lines 226-238).

Main comment 2.2

"Secondly, I am unclear why the authors did not consider the inclusion of Ta, e.g., via. Nb/Ta? I would have thought that there would be sufficient high-quality data for TTG and non-TTG granitoids to test whether Nb/Ta instead of [Nb] would yield even more discrimination power? At the very least, more context needs to be given to Nb/Ta, as rutile and ilmenite have different Ds not just for Nb but for Ta also, i.e., the foundation of the Hoffmann et al. (2011) model, which is cited as [19] but not really discussed in a meaningful way, i.e. as a testable hypothesis."

We looked into using Nb/Ta, both instead of Nb concentrations in the petrogenetic interpretation of rutile vs. ilmenite in the source as well as in the general context of partial melting, assimilation and fractional crystallization. The issues that we encountered are two-fold: 1) the partition coefficients for Nb and Ta in various phases are sensitive to pressure and H₂O concentration, which are difficult to constrain precisely for TTG melting; 2) the data for Nb/Ta in TTGs are generally not precise. Hoffmann et al. (2011) are among few who constrain(ed) Nb and Ta concentrations (and thus Nb/Ta) using isotope dilution. Most trace-element data available for TTGs include Ta data obtained using "normal" internally standardized trace-element analysis, which is limited by low count statistics on the m/z corresponding to ¹⁸¹Ta. The resulting Ta concentration and Nb/Ta data are associated with a degree of scatter that is notably more significant than that of the concentration data obtained for the more abundant Nb. The combined result of these issues is that uncertainties on the partition coefficients limits the use of Nb/Ta in petrogenetic modelling, whereas the empirical Nb/Ta data are mostly not precise enough to test any results that such models would provide. Recognizing these issues, we decided to focus on Nb, which is more abundant than the chemically similar Ta (see new note in lines 81-86) and provides a relatively low-dispersion parameter for investigating HFSE partitioning during melting. We do agree that Hoffmann et al. (2011) could have been used more effectively, as the new data indeed allow (re-)interpretation of data in its context. This is now explained in the discussion.

→ New text was added in lines 312-319.

Main comment 2.3

"Thirdly, I was surprised to see the authors opt to calculate Ti relative to Zr and Eu. I can understand the inclusion of Zr instead of a MREE but am unclear why Eu was preferred over Gd, as this choice now 'marries' the effects of Ti-phases and plagioclase. I think a revised submission would need to provide a diagram equivalent to Figure 1 with Ti_N/Ti* calculated with Ti* as sqrt(Gd_N × Zr_N) so that readers (and reviewers) can scrutinise the authors' claim that their preferred Ti* is indeed superior."*

This is a very useful comment. We followed one of the conventions regarding incompatibility, which brackets Ti with Zr and Eu. We, however, do appreciate that Eu comes may be problematic due to additional effects from plagioclase. As advised, we replaced Eu with Gd, and thus now define the Ti anomaly as Ti_N/Ti* with Ti* being $\sqrt{Zr_N \times Gd_N}$. All calculations were redone with this new variable and all figures were changed accordingly. The result is better than with Eu, most likely because Gd is not associated with the dispersion that is built into Eu due to additional processes affecting this element.

→ Changes were made to the text (lines 90-92), all calculations were redone with the new variable, and all figures were modified accordingly.

REVIEWERS' COMMENTS

Reviewer #1 (Remarks to the Author):

This is a much improved manuscript. I am broadly satisfied with the revision. Please find some minor comments below for your consideration.

Line 102-103: DNb for which phase and Dilm for which element? Please clarify these. I suggest the authors using superscript for phases and subscript for element, such as DN^bilm.

Line 106: "Assimilation of material containing such phases," Please clarify the phases.

Line 162-163: In my eyes, the first feeling of the trace elements of the low F TTGs is that large amount of plagioclase in the mineral assemblage. In the paper of Kendrick et al. (2022, Geology), the Group 3 TTGs are suggested to be resulted from plagioclase accumulation, these TTGs have high (TiN/Ti*)/Nb, and all the trace element are consistent with the low F TTGs. How to explain this? They have the field, petrological and modelling evidence.

Line 169-170: Based on the calculation (Fig. 3), more than half of the TTGs experience only 10% of fractional crystallization. If it is true, then TTGs with low F would contain significant amount of amphibole. However, amphibole is scarce in TTGs, as summarized by Moyen 2012.

Line 194: I believe the plagioclase-cumulate source rocks would produce TTG like melt compositions, especially the HP TTGs. It should be a potential protolith. However, I suspect that all the TTGs were primarily produced through partial melting of plagioclase-cumulate source rock. If so, what will basalts (much more abundant and fertile rocks) produce.

Line 247-249: Water contents in the amphibole are around 2.5 wt%. Then the rocks with 90 vol% of amphibole would have water content of 2.5 wt%. What else hydrous minerals could have higher water content, biotite?

Line 303: I think the authors made a mistake here. Is it (TiN/Ti*)/Nb?

Line 320: If this is the reason for sanukitoid production, sanukitoids would be ubiquitous in the early earth. TTGs with age older than 3.0 Ga could be seen in almost every craton. But why sanukitoids older than 3.0 Ga is scarce?

Reviewer #2 (Remarks to the Author):

Having reviewed the original submission and seen the author's responses to both reviews, I am in general satisfied that this m/s can now be considered for publication. I think that the authors addressed by points #1 and #3 very well and I am pleased that my suggestion to calculate Nb* via Gd turned out to be constructive and useful. I am not sold that 'ordinary' ICP-MS analysis is unable to generate high precision Nb/Ta data but as long as no such claim is made in the m/s, I'm ok with Nb/Ta remaining unexplored (and available to be visited in future studies). I do get the point about the Ds for Nb and Ta being quite sensitive to factors that were not the main targets of this study.

In terms of further improving the m/s, I would like the authors to consider a bit more deeply, the geodynamic setting of the envisaged melting to make TTGs. The longer I am pondering this topic, the harder I find it to see TTGs as liquids from a closed system. Since reviewing the m/s, some of my thoughts on the topic have been published (pages 15-21 of the article below). I'm not insisting on getting cited in this m/s, but I think the authors may be well advised considering hybridisation of the following kind:

gabbro cumulate + picrite = new solids - (melted amphibole + garnet) + silicic liquid (=TTG) as a geodynamic setting at the base of Archaean crust.

REVIEWER

Comment number

Original comment or suggestion.

Response to comment.

→ Summary of concrete changes made with line numbers (where necessary).

REVIEWER 1

Minor comment 1.1

"Line 102-103: D_{Nb} for which phase and D^{ilm} for which element? Please clarify these. I suggest the authors using superscript for phases and subscript for element, such as D_{Nb}^{ilm} ."

The terms in this sentence were indeed incorrect and the syntax was also not correct. The sentence was corrected. We do use  for elements and  for phases otherwise.

→ The revised sentence is in lines 102-104.

Minor comment 1.2

"Line 106: "Assimilation of material containing such phases," Please clarify the phases."

The phases are now specified.

→ The revised sentence is in lines 107-108.

Minor comment 1.3

"In my eyes, the first feeling of the trace elements of the low F TTGs is that large amount of plagioclase in the mineral assemblage. In the paper of Kendrick et al. (2022, Geology), the Group 3 TTGs are suggested to be resulted from plagioclase accumulation, these TTGs have high $(TiN/Ti^)/Nb$, and all the trace element are consistent with the low F TTGs. How to explain this? They have the field, petrological and modelling evidence."*

We agree that plagioclase accumulation likely plays a significant role in TTG melt evolution and we note this as well in the text (lines 177-179, 186-200). This process nevertheless does not adequately explain the composition of high-

(Ti_N/Ti*)/Nb TTGs, as it is inconsistent with the major-element compositions and compositional trends observed among this particular subset of samples. This is now explained in new short section of text, which was developed to ensure this point is clear.

→ The revised sentence is in lines 143-150.

Minor comment 1.4

"Line 169-170: Based on the calculation (Fig. 3), more than half of the TTGs experience only 10% of fractional crystallization. If it is true, then TTGs with low F would contain significant amount of amphibole. However, amphibole is scarce in TTGs, as summarized by Moyen 2012."

We agree that modal abundances of amphibole in TTGs is generally low and have noted this in the discussion (lines XXX). In fact, this is an important observation that is consistent, rather than inconsistent, with amphibole fractional crystallization having affected most TTGs (see also Rollinson, 2021). The dearth of amphibole-rich TTGs may simply reflect the dearth in low-*F* TTGs, which indeed make up only a small fraction of TTGs sampled and analyzed so far.

On the basis of the proposed model, a simple relationship may be predicted between amphibole modal abundance and *F*, with low-*F* having the highest abundance. Testing this prediction is nevertheless complicated in two ways: 1) Mineral modal abundances are not available for the TTGs compiled so far and also cannot be deduced from the data; CIPW-normative mineral abundance calculations do not account for hydrous minerals and measured H₂O concentrations, which may provide an avenue to calculate amp/cpx in a modified normative scheme, are impacted by processes that happened after any amphibole fractionations has occurred (devolatilization, fluid-rock interaction). 2) It can never really be ruled out that a melt that at some point lost amphibole by fractional crystallization later accumulated some. Quantitative assessments in this regard cannot be made given that other minerals could also affect melt compositions in the process; this impedes quantifying "net amphibole loss".

A proper empirical test with regards to the role of amphibole fractional crystallization would require samples with different *F* from a single TTG pluton for which it can be positively ruled out that any process other than amphibole fractionation has affected magma compositions. Such test may be possible and would be valuable. However, as this is yet to be done, we: 1) rely on experimental partition coefficients to predict the effects of fractional crystallization, and 2) remain conservative and treat *F* as a relative measure of crystal fractionation (lines 161-163).

The key observations for our study are that the modal abundance of amphibole is generally low and that hornblende cumulates coexist with TTGs as complementary reservoir (lines 186-200). We consider these observations to provide sufficient supporting evidence for our interpretations of *F* and TTG compositions in general.

Minor comment 1.5

"Line 194: I believe the plagioclase-cumulate source rocks would produce TTG like melt compositions, especially the HP TTGs. It should be a potential protolith. However, I suspect that all the TTGs were primarily produced through partial melting of plagioclase-cumulate source rock. If so, what will basalts (much more abundant and fertile rocks) produce."

Basaltic source rocks can be ruled out and the reasons for this are explained (melts from basaltic sources would not produce the high La/Sm and Sm/Yb, and low Eu_N/Eu* as seen in TTGs). To make this clearer, we added "basaltic" in the specific sentence where we discuss this aspect. This small change ensures accuracy and clarity of text.

→ The revised sentence is in lines 224-228.

Minor comment 1.6

"Line 247-249: Water contents in the amphibole are around 2.5 wt%. Then the rocks with 90 vol% of amphibole would have water content of 2.5 wt%. What else hydrous minerals could have higher water content, biotite?"

This is an interesting aspect. Amphibole is likely the main crystalline repository for H₂O in gabbroic rocks; nominally anhydrous minerals, such as clinopyroxene or olivine, may contain small amounts of H₂O as well. Biotite

is not commonly present in significant modal abundance. What does make gabbroic rocks hydrous, or at least more hydrous than is commonly thought, is fluid inclusions. We consider the H₂O component to gabbros an interesting aspect to the point that we had quite a few sentences on this in one of the original drafts. The text was nevertheless speculative and somewhat beyond-the-scope, and thus was omitted. We realize that at least some detail in this regard may be useful and have thus slightly enhanced the sentence discussing this aspect.

→ The revised sentence is in lines 262-265.

Minor comment 1.7

"Line 303: I think the authors made a mistake here. Is it (TiN/Ti)/Nb?"*

The reviewer is correct! Important catch. The ratio is corrected.

→ The revised sentence is in lines 329-321.

Minor comment 1.8

"Line 320: If this is the reason for sanukitoid production, sanukitoids would be ubiquitous in the early earth. TTGs with age older than 3.0 Ga could be seen in almost every craton. But why sanukitoids older than 3.0 Ga is scarce?"

Sanukitoids are not common in any Archean craton and their apparent occurrence since 3 Gyr is likely a preservation issue. TTGs younger than 3 Gyr are more commonly preserved and sampled, and thus many of their more unusual features will appear to be more common in, if not restricted to, the Meso- and Neoproterozoic. Older sanukitoids may have become lost during overprinting, given that a tectonically reworked sanukitoid can easily be mistaken for an ordinary granitic gneiss.

Going into the preservation vs. occurrence discussion is a bit too tangential to our manuscript, but we do recognize that we did not properly emphasize the rarity and sporadic occurrence of sanukitoids (or their potential modern-day equivalents). The text indeed read as though sanukitoids *always* are part of this causal chain, which is likely what the reviewer is alluding to. To address this issue, we made a few small changes to better communicate scarcity and preservation bias.

→ The revised sentences are in lines 350-352 and 354-356 (about the rarity of such rocks in modern settings), and lines 360-361 (about rarity and preservation issues).

REVIEWER 2:

Minor comment

"In terms of further improving the m/s, I would like the authors to consider a bit more deeply, the geodynamic setting of the envisaged melting to make TTGs. The longer I am pondering this topic, the harder I find it to see TTGs as liquids from a closed system. Since reviewing the m/s, some of my thoughts on the topic have been published (pages 15-21 of the article below). I'm not insisting on getting cited in this m/s, but I think the authors may be well advised considering hybridisation of the following kind: gabbro cumulate + picrite = new solids - (melted amphibole + garnet) + silicic liquid (=TTG) as a geodynamic setting at the base of Archean crust."

We did consider such hybridized rocks as a possible "gabbro-like" source lithology in the revised version of our manuscript. We do, however, appreciate that this may not have been clear and could have been phrased more precisely. The sentence where this was noted is improved. The Kamber & Ossa Ossa paper is a wonderful review paper and it provides the perfect context for our findings. The content that is of particular relevance to our manuscript is nevertheless already covered in the seminal Kamber (2015) review and in Emo & Kamber (2022), both of which are cited. We consider those sufficient to provide the literature background for the points made.

→ The revised sentence is in lines 256-258.